# FedMT: Federated Learning with Mixed-type Labels

## Abstract

In *federated learning* (FL), classifiers (e.g., deep networks) are trained on datasets from multiple centers without exchanging data across them, and thus improves *sample efficiency*. In the classical setting of FL, the *same labeling criterion* is usually employed across all centers being involved in training. This constraint greatly limits the applicability of FL. For example, standards used for *disease diagnosis* are more likely to be different across clinical centers, which mismatches the classical FL setting. In this paper, we consider an important yet under-explored setting of FL, namely FL with *mixed-type labels* where *different labeling criteria* can be employed by various centers, leading to inter-center label space differences and challenging existing FL methods designed for the classical setting. To effectively and efficiently train models with mixed-type labels, we propose a theory-guided and model-agnostic approach that can make use of the underlying correspondence between those label spaces and can be easily combined with various FL methods such as FedAvg. We present *convergence analysis* based on over-parameterized ReLU networks. We show that the proposed method can achieve linear convergence in label projection, and demonstrate the impact of the parameters of our new setting on the convergence rate. The proposed method is evaluated and the theoretical findings are validated on benchmark and medical datasets.

## 1 Introduction

Federated learning (FL) enables centers to jointly learn a model while keeping data at each center. It avoids the centralization of data which is restricted by regulations such as CCPA (Legislature, 2018), HIPAA (Act, 1996), and GDPR (Voigt et al., 2018) and has gained popularity in various applications. The widely used FL methods, such as FedAvg (McMahan et al., 2017), FedAdam (Reddi et al., 2020), and others use iterative optimization algorithms to achieve jointly model training across centers. At each round, local center performs stochastic gradient descent (SGD) for several steps then centers communicate their current model weight to a central server to be aggregated.

When training a classifier in the classical FL setting, the datasets across all centers are annotated with the same labeling criterion. However, in real applications such as healthcare, standards for disease diagnosis may be different across clinical centers due to varying levels of expertise or technology available at different sites. For example, when diagnosing ADHD with brain imaging, the labels are usually acquired over a long period of behavior studies. Different centers may follow different diagnosis and statistical manuals (McKeown et al., 2015) and it is difficult to ask centers to relabel data using a unified criterion as some behavior studies cannot be repeated. This leads to different label spaces across centers. In addition, the center with the most complex labeling criterion, whose label space is desired for future prediction, typically only has limited labeled samples due to labeling difficulty or cost. In this paper, we aim to answer the following important question:

*With limited samples from the desired label space, how to leverage the commonly used FL pipeline (e.g., FedAvg) and data from other centers in different label spaces to jointly learn an FL model in the desired label space, without additional feature exchanging and data relabeling?*

**Problem Setting:** We study an FL problem for a given classification task. Each center has one labeling criterion, and the criteria across centers can be different. Samples do not overlap across centers. As shown in Fig. 1, first, label spaces are **not** necessarily nested. One class from the desired

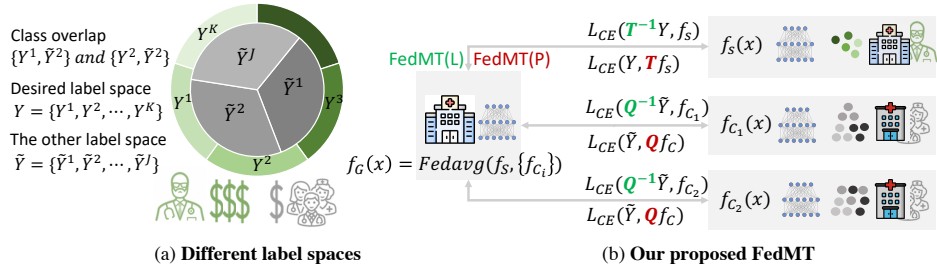

(a) **Different label spaces**  (b) **Our proposed FedMT**

Figure 1: Illustration of the problem setting and our proposed `FedMT` method. (a) We consider different label spaces (*i.e.,* desired label space $\mathcal{Y}$ with $K$ classes and the other space $\widetilde{\mathcal{Y}}$ with $J$ classes) where classes may overlap, such as $Y^1$ and $\widetilde{Y}^2$. Annotation using the desired label criterion is usually harder and more expensive to obtain, thus less such labeled samples are available. (b) We use fixed label space correspondence matrix $\boldsymbol{Q}$ to associate label spaces $\widetilde{\mathcal{Y}}$ with $\mathcal{Y}$ and noise correction matrix $\boldsymbol{T}$ to correct label noises in $\mathcal{Y}$ (if any). We correct predictions by multiplying classifiers' probability output $f$ by projection matrices locally (`FedMT` (P)) or correct labels by multiplying by the inverse of projection matrices to sample observed labels (`FedMT` (L)) under FedAvg framework.

space may overlaps with different classes in another space and vice versa (*e.g.,* , disease diagnoses often exhibit imperfect agreement). Second, following the motivated healthcare example, we assume *limited* amount of labeled data ($< 5\%$) in the *desired label space* is available [1]. Moreover, for the ease of experiment design, we consider the case where these data are stored in one '*specialized center*,' and this center can be treated as the server to coordinate FL but still perform local model updating like the other clients, *i.e.,* the centers with the other labeling criteria. All the centers jointly train an FL model following the standard FL training protocol as shown in Fig. 1 (b).

Prior methods for dealing with different labels spaces include personalized FL (Collins et al., 2021), but they fail to leverage the correspondence across different label spaces. Transfer learning (Yang et al., 2019) which pretrains a model on one space and finetunes the pretrained model on other spaces can be an alternative solution in FL, but sub-optimal pretraining may lead to negative transfer (Chen et al., 2019). Therefore, to address the limitation of above methods, we want to a) simultaneously leverage different types of labels and their correspondence and b) learn FL model end-to-end. To the best of our knowledge, other possible centralized methods meet our needs a) and b) are restricted to coarse to fine label spaces that have hierarchical structures (Touvron et al., 2021; Chen et al., 2021a), which does not hold for the general problem of our interest, or require pulling all data features together for similarity comparison using more sophisticated training strategies (Hu et al., 2022). These methods cannot be simply extended to widely used FL methods (*e.g.,* FedAvg) and require feature sharing across centers which increases privacy risks.

To address the above limitations, we propose a plug-and-play method called `FedMT`, which is a versatile strategy that can be easily combined with various FL pipelines such as FedAvg. Specifically, we use models with the same architecture whose output dimension is the number of classes in the desired label space across all centers. To use client data from the other label space for supervision, we align two spaces either with label (probability) projection that projects the label (class scores) to the other space. We further show that our methods has the bonus to handle label noise.

**Contributions:** Our contributions are three folds. *Methodologically*, we propose a novel FL method, `FedMT`, which is a computationally efficient and versatile solution *Theoretically*, we present the convergence of `FedMT` in FL with over-parameterized ReLU neural networks, and explore the impact of amount of data from desired label space and different noise levels; *Empirically*, we demonstrate the superior results in this challenging setting over prior art with extensive experiments on benchmark and medical datasets.

## 2   RELATED WORK

**Federated learning** FL is emerging as a learning paradigm for distributed clients that bypasses data sharing to train local models collaboratively. To aggregate model parameters, FedAvg (McMahan

---

[1]Due to labeling difficulties, such labels can also be noisy. Hence, we also explore this property in our work.

et al., 2017) is the most widely used approach in FL. Variants of FedAvg have been proposed to improve optimization (Reddi et al., 2020; Rothchild et al., 2020) and for non-iid data (Li et al., 2021; 2020a; Karimireddy et al., 2020). Recently FL methods have been studied in semi-supervised learning (Jeong et al., 2021; Bdair et al., 2021), weakly supervised learning (Lu et al., 2021), and positive label only learning (Yu et al., 2020). Theoretical studies of FL have not yet directly addressed neural network algorithms nor accounted for the influence of individual data samples (Karimireddy et al., 2020; Li et al., 2019; Khaled et al., 2020). Recently, a theoretical framework for FL on fully supervised neural network regression using FedAvg was proposed in Huang et al. (2021). To the best of our knowledge, this is the first work to investigate FL's theory on neural network classification under mixed-type labeled data.

**Neural Tangent Kernel (NTK)** NTK is an essential tool to study the learnability of neural networks. NTK was first studied by Jacot et al. (2018) showing the equivalence between training an infinitely wide neural network with gradient descent and kernel regression. NTK theory has been extended to convolutional neural networks (CNNs) (Arora et al., 2019), graph neural networks (GNNs) (Du et al., 2019; Jiang et al., 2019), recurrent neural networks (RNNs) (Alemohammad et al., 2020). NTK for FL was recently studied by Huang et al. (2021), in the supervised FL regression problem using mean squared error loss.

## 3 METHODS

### 3.1 PRELIMINARIES: CLASSICAL FL

We start by reviewing the classical FL approach, FedAvg, with full supervision (McMahan et al., 2017) for classification with the same labeling criterion. We consider a $K$-class classification problem with the feature space $\mathcal{X}$ and the label space $\mathcal{Y} = [K]$. Let $\boldsymbol{x} \in \mathcal{X}$ and $y \in \mathcal{Y}$ be the input and output random variables following an underlying joint distribution with density $p(\boldsymbol{x}, y)$. Let $\boldsymbol{f} : \mathcal{X} \to \mathbb{R}^K$ be a $K$-class classifier such that $\sigma(f_k(\boldsymbol{x})) = p(y = k|\boldsymbol{x})$, where $f_k(\boldsymbol{x})$ is the $k$-th element of $\boldsymbol{f}(\boldsymbol{x})$ and $\sigma(f_k) = \exp(f_k)/\sum_{k'=1}^K \exp(f_{k'})$ is the softmax function. Then the predicted label can be obtained via $y_{\text{pred}} = \arg\max_{k \in [K]} f_k(\boldsymbol{x})$.

In the classical FL setup, each client $c \in [C]$ has access to a labeled training set $\mathcal{D}_c = \{(\boldsymbol{x}_i^c, y_i^c)\}_{i=1}^{N_c}$ of size $N_c$ and learns its local model $\boldsymbol{f}^c$ by minimizing the following empirical risk:

$$\widehat{R}_c^{\text{a}}(\boldsymbol{f}^c; \mathcal{D}_c) = \frac{1}{N_c} \sum_{i=1}^{N_c} \ell(\boldsymbol{f}^c(\boldsymbol{x}_i^c), y_i^c), \tag{1}$$

where $\ell$ is the *cross-entropy loss*, i.e., $\ell_{\text{CE}}(\boldsymbol{f}^c(\boldsymbol{x}), y) = -\sum_{k=1}^K \mathbb{1}(y = k) \log f_k^c(\boldsymbol{x}) = -\log f_y^c(\boldsymbol{x})$ and $\mathbb{1}(\cdot)$ is the indicator function. The goal of classical FL is that $C$ clients collaboratively train a global classification model $\boldsymbol{f}$ that generalizes well with respect to $p(\boldsymbol{x}, y)$, without sharing their local data $\mathcal{D}_c$. The problem can be formalized as minimizing the aggregated risk: $R(\boldsymbol{f}) = \frac{1}{C} \sum_{c=1}^C \widehat{R}_c^{\text{a}}(\boldsymbol{f}; \mathcal{D}_c)$. FedAvg employs a server to coordinate the iterative distributed training through model parameter averaging as described in Algorithm 1.

### 3.2 PROBLEM FORMULATION

In contrast to the classical supervised FL classification, where all the centers share the same label space, we consider the case where the label spaces across centers may be different. For the ease of presentation, we assume there are two label spaces, namely $\mathcal{Y} = \{Y^k\}_{k=1}^K$ and $\widetilde{\mathcal{Y}} = \{\widetilde{Y}^j\}_{j=1}^J$ where $K > J$. [2] For the 'specialized center' with the desired label space, we denote it as a server and its dataset as $\mathcal{D}^s = \{(\boldsymbol{x}_i^s, y_i^s) : i \in [nK]\}$, where $y_i^s \in \mathcal{Y}$. We also have in total $N$ labeled data with a different criterion that are stored across $C$ clients. For $c \in [C]$, we denote by $S_c$ the indices of data from the other label space $\mathcal{D}^c = \{(\boldsymbol{x}_i^c, y_i^c) : i \in S_c\}$ on $c$-th client, where $y_i^c \in \widetilde{\mathcal{Y}}$. Let $N_c = |S_c|$ and we have $N = \sum_{c \in [C]} N_c$. Our objective is to train a global classifier in the desired label space $\boldsymbol{f} : \mathcal{X} \mapsto \mathcal{Y}$ using data in the system from different label spaces.

---

[2]Our method can be easily extended to multiple label spaces.

### 3.3 PROPOSED METHOD

**Motivation** Learning under different label spaces can be formulated as a corrupted label learning problem (Van Rooyen & Williamson, 2017; Patrini et al., 2017), where label space with less classes $\widetilde{\mathcal{Y}}$ is treated as the corrupted observation of a true underlying label space $\mathcal{Y}$ with more classes, *i.e.,* our desired label space. Our method is based on loss correction (Van Rooyen & Williamson, 2017; Patrini et al., 2017), which is a well-established corrupted label learning approach in the centralized domain with statistically consistency guarantees.

Let $\mathbb{T} : \mathcal{Y} \mapsto \widetilde{\mathcal{Y}}$ be a linear transformation, whose pseudo inverse is $\mathbb{T}^{-1}$. Under proper assumptions (see Theorem 3.1) and a given function $\boldsymbol{f} : \mathcal{X} \mapsto \mathcal{Y}$, Patrini et al. (2017) showed two ways to perform loss correction that $\mathbb{T}^{-1}$ can pull back from functions of corrupted labels to functions of true labels and $\mathbb{T}$ can transfer functions of true labels to those of corrupted labels. Mathematically, this can be written as:

**Theorem 3.1** (Loss correction (informal) (Patrini et al., 2017; Van Rooyen & Williamson, 2017)). *Given a non-singular linear mapping matrix $\mathbb{T}$ and a proper loss $\ell$, one can achieve the same minimizer of the original loss under the true label distribution:*

$$\arg\min_{\boldsymbol{f}} \mathbb{E}_{\boldsymbol{x},\boldsymbol{y} \in \mathcal{Y}} \ell(\boldsymbol{y}, \boldsymbol{f}) = \arg\min_{\boldsymbol{f}} \mathbb{E}_{\boldsymbol{x},\boldsymbol{y} \in \widetilde{\mathcal{Y}}} \ell(\boldsymbol{y}, \mathbb{T}\boldsymbol{f}) \quad and \quad \mathbb{E}_{\boldsymbol{x},\boldsymbol{y} \in \mathcal{Y}} \ell(\boldsymbol{y}, \boldsymbol{f}) = \mathbb{E}_{\boldsymbol{x},\boldsymbol{y} \in \widetilde{\mathcal{Y}}} \mathbb{T}^{-1} \ell(\boldsymbol{y}, \boldsymbol{f}).$$

In addition, loss correction methods are model agnostic and can be generalized to many loss functions, therefore provide a versatile framework for learning from corrupted labels. Since our goal is to jointly learn a classifier $\boldsymbol{f} : \mathcal{X} \to \mathbb{R}^K$ for clients and server, the overall loss can be written as:

$$R_{\text{overall}}(\boldsymbol{f}) = \frac{1}{C+1} \left\{ \widehat{R}_s(\boldsymbol{f}) + \sum_{c=1}^{C} \widehat{R}_c(\boldsymbol{f}; \mathcal{D}^c) \right\}, \tag{2}$$

where $\widehat{R}_s(\boldsymbol{f})$ and $\widehat{R}_c(\boldsymbol{f})$ are respectively the empirical risk based on the server's data and the clients' data from a different label space. To minimize the overall loss in (2), we need to align the predictions or losses of both types of labels to the underlying true label space. Considering the specific form of cross-entropy (1), we adapt the findings of Patrini et al. (2017); Van Rooyen & Williamson (2017) to our FL setting. For all the clients in FL $l \in [C]$, we propose leveraging the following two kinds of projections to both server and clients as:

**Probability projection**: $\widehat{R}_l(\boldsymbol{f}; \mathcal{D}^l) = -\frac{1}{N_l} \sum_{i \in S_l} \sum_{j=1}^{J} \mathbb{1}(y_i = \widetilde{Y}^j) \log \left\{ \sum_{k=1}^{K} \mathbb{T}_{jk}^l \sigma(f_k(\boldsymbol{x}_i)) \right\}, \tag{3}$

**Label projection**: $\widehat{R}_l(\boldsymbol{f}; \mathcal{D}^l) = -\frac{1}{N_l} \sum_{i \in S_l} \sum_{k=1}^{K} \left\{ \sum_{j=1}^{J} \mathbb{T}_{kj}^{-1} \mathbb{1}(y_i = \widetilde{Y}^j) \right\} \log \sigma(f_k(\boldsymbol{x}_i)). \tag{4}$

For clients, we denote $\mathbb{T} = \boldsymbol{Q} \in [0,1]^{J \times K}$, whose $j$-th row denotes the mixing weights of classes in the desired label space for $j$-th class in the other label space. As both probability projection and label projection are performed locally, the overall loss (2) can be optimized using general FL strategies, such as FedAvg (McMahan et al., 2017) used in this work, or other variants like Li et al. (2020a; 2021); Karimireddy et al. (2020). Note our method can be easily extended to more than one centers in the desired label space as shown in Appendix D.5.

**Extension to Noisy Labels** Another advantage of our method is that it has the flexibility to extend to the case where there are noisy observations, that is the label is mislabeled to a wrong one in the same label space. Let us explain with the case where the server has noisy labels. Specifically, on the server, we let $\mathbb{T} = \boldsymbol{T} \in [0,1]^{K \times K}$ be a matrix whose $(i,j)$-th element denotes $p(\widetilde{y}^s = Y^j | y^s = Y^i)$ where $\widetilde{y}^s$ is the observed noisy label and $y^s$ is the true label.

The detailed FedMT algorithm is described in Algorithm 2, $\boldsymbol{T} \neq \mathbf{I}^{K \times K}$ under noisy label case and $\boldsymbol{T} = \mathbf{I}^{K \times K}$ for noise-free setting. In this algorithm, it is worth noting that FedMT is easy to implement as it only slightly modifies FedAvg as highlighted in blue.

**Algorithm 1** FL using FedAvg (McMahan et al., 2017)

**Server Input:** initial $\boldsymbol{f}$, aggregation step-size $\eta_{\text{agg}}$, and global communication round $R$

**Client Input:** local model $\boldsymbol{f}^c$, local dataset $\mathcal{D}_c$, SGD step-size $\eta_{\text{sgd}}$, and local updating iterations $t$ (for $c \in [C]$)

1: For $r = 1 \to R$ rounds, we run **A** on each client and **B** iteratively .
2: **procedure A**. MODELUPDATE($r$)
3:    $\boldsymbol{f}^c \leftarrow \boldsymbol{f}$    ▷ Receive updated model from PROC. B
4:    **for** $\tau = 1 \to t$ **do**
5:        $\boldsymbol{f}^c \leftarrow \boldsymbol{f}^c - \eta_{\text{sgd}} \cdot \nabla \widehat{R}_c^{\text{a}}(\boldsymbol{f}^c; \mathcal{D}_c)$
      ▷ Model updates via SGD
6:    send $\boldsymbol{f}^c - \boldsymbol{f}$ to PROC. B
7: **procedure B**. MODELAGG($r$)
8:    receive model updates $\boldsymbol{f}^c - \boldsymbol{f}$ from PROC. A
9:    $\boldsymbol{f} \leftarrow \boldsymbol{f} - \eta_{\text{agg}} \cdot \sum_{c=1}^{C}(\boldsymbol{f}^c - \boldsymbol{f})$
10:    broadcast $\boldsymbol{f}$ to PROC. A

**Algorithm 2** FL using FedMT (Ours)

**Server Input:** inputs of Alg 1, server model $\boldsymbol{f}^s$, *small* noisy dataset $\mathcal{D}_s = \{\boldsymbol{x}^s, \boldsymbol{y}^s\}$ where $\boldsymbol{y}^s \in \mathcal{Y}$, projection matrix $T$

**Client Input:** inputs of Alg 1 but $\mathcal{D}_c = \{\boldsymbol{x}^c, \boldsymbol{y}^c\}$ where $\boldsymbol{y}^c \in \widetilde{\mathcal{Y}}$, and projection matrices $\boldsymbol{Q}$

1: For $r = 1 \to R$ rounds, we run **A** on each client and **B** iteratively .
2: **procedure A**.MODELUPDATE($r$)
3:    $\boldsymbol{f}^c \leftarrow \boldsymbol{f}$   ▷ Receive updated model from PROC. B
4:    **for** $\tau = 1 \to t$ **do**
5:        **if** probability projection **then**
6:            $\boldsymbol{f}^c \leftarrow \boldsymbol{f}^c - \eta_{\text{sgd}} \cdot \nabla \widehat{R}_c(\boldsymbol{Q}\boldsymbol{f}^c; \boldsymbol{y}^c)$
7:            $\boldsymbol{f}^s \leftarrow \boldsymbol{f}^s - \eta_{\text{sgd}} \cdot \nabla \widehat{R}_s(\boldsymbol{T}\boldsymbol{f}^s; \boldsymbol{y}^s)$
8:        **else if** label projection **then**
9:            $\boldsymbol{f}^c \leftarrow \boldsymbol{f}^c - \eta_{\text{sgd}} \cdot \nabla \widehat{R}_c(\boldsymbol{f}^c; \boldsymbol{Q}^{-1}\boldsymbol{y}^c)$
10:            $\boldsymbol{f}^s \leftarrow \boldsymbol{f}^s - \eta_{\text{sgd}} \cdot \nabla \widehat{R}_s(\boldsymbol{f}^s; \boldsymbol{T}^{-1}\boldsymbol{y}^s)$
11:    send $\boldsymbol{f}^l - \boldsymbol{f}$ to PROC. B for $l \in \{[C], s\}$
12: **procedure B**. MODELAGG($r$)
13:    receive model updates from PROC. A
14:    $\boldsymbol{f} \leftarrow \boldsymbol{f} - \eta_{\text{agg}} \cdot \sum_{l \in \{[C], s\}}(\boldsymbol{f}^l - \boldsymbol{f})$
15:    broadcast $\boldsymbol{f}$ to PROC. A

## 3.4 THEORETICAL ANALYSIS

Although label projection-based loss correction methods have shown to have good properties under centralized setting (see Theorem 3.1 Patrini et al. (2017); Van Rooyen & Williamson (2017)), their theoretical convergence under FL have not been explored. Hence, we focus on the theoretical analysis of label projection in (4), given it has been shown to have better theoretical guarantees than (3) in a previous study (see Theorem 3.1 (Patrini et al., 2017)). Further, we are interested in investigating the impact of various parameters on theoretical convergence. In this section, we establish a novel theoretical analysis of FedMT using NTK (Arora et al., 2019; Lee et al., 2019; Huang et al., 2021).

**NTK setup**    Our theoretical results are studied under an over-parameterized one-hidden layer neural network (NN). Let $\boldsymbol{f} : \mathcal{X} \to \mathbb{R}^K$ be the output of the NN whose $k$th element is

$$f_k(\boldsymbol{u}, \boldsymbol{x}) = \frac{1}{\sqrt{M}} \sum_{m=1}^{M} a_{km} \phi(\boldsymbol{u}_m^\top \boldsymbol{x})$$

where $\phi(z) = \max\{z, 0\}$ is the ReLU activation function and $\boldsymbol{u} = [\boldsymbol{u}_1, \boldsymbol{u}_2, \ldots, \boldsymbol{u}_M] \in \mathbb{R}^{d \times M}$.

**Definition 3.2** (Initialization). *We initialize $\boldsymbol{u} \in \mathbb{R}^{d \times M}$ and $a_{km}$ as follows. For each $m \in [M]$, $\boldsymbol{u}_m$ is sampled from $\mathcal{N}(0, I)$. For each $k \in [K]$ and $m \in [M]$, $a_{km}$ is sampled from $\{-1, +1\}$ uniformly at random and is not trainable.*

In the $r$-th global round, the server broadcasts the global model weight $\boldsymbol{u}_m(r)$ to every client. Each client $c$ then starts from $\boldsymbol{u}_{m,c}(0, r) = \boldsymbol{u}(r)$ and takes $t$ local gradient steps via gradient descent with step size $\eta_{\text{local}}$

$$\boldsymbol{u}_{m,c}(\tau + 1, r) = \boldsymbol{u}_{m,c}(\tau, r) - \eta_{\text{local}} \frac{\partial \widehat{R}_c(\boldsymbol{u}_{m,c}(\tau, r))}{\partial \boldsymbol{u}_m}$$

where $\boldsymbol{u}_{m,c}(\tau, r)$ is the value of $\boldsymbol{u}_m$ on $c$th client at step $\tau$ in $r$-th global round. Then the client sends $\Delta \boldsymbol{u}_{m,c}(r) = \boldsymbol{u}_{m,c}(t, r) - \boldsymbol{u}_{m,c}(0, r)$ to server and server computes a new $\boldsymbol{u}_{m,c}(r + 1)$ based on the average of all $\Delta \boldsymbol{u}_{m,c}(r)$ via

$$\boldsymbol{u}_m(r + 1) = \boldsymbol{u}_m(r) + \eta_{\text{agg}} \cdot \sum_{c \in [C]} \Delta \boldsymbol{u}_{m,c}(r) / C.$$

**NTK analysis results** Let $g(\boldsymbol{u}, \boldsymbol{x}) = \sigma(\boldsymbol{f}(\boldsymbol{u}, \boldsymbol{x}))$ and $\boldsymbol{y}_i^c$ be the multi-hot vector representation of $y_i^c$ and $\boldsymbol{y}_i^s$ be the one-hot vector representation of $y_i^s$ in $\mathbb{R}^K$. Let $\boldsymbol{Q}$ be the linear mapping from $\mathcal{Y}$ to $\widetilde{\mathcal{Y}}$ and $\boldsymbol{T}$ be the same as before. Following Neural Tangent Kernel (NTK) analysis (Lee et al., 2019; Arora et al., 2019), we consider the mean squared error loss[3] respectively on the client and server

$$\widehat{R}_c(\boldsymbol{u}) = \frac{1}{N_c} \sum_{i \in S_c} \sum_{j=1}^J \sum_{k=1}^K Q_{jk}^{-1} (y_{ik}^c - g_k(\boldsymbol{u}, \boldsymbol{x}_i^c))^2,$$

$$\widehat{R}_s(\boldsymbol{u}) = \frac{1}{nK} \sum_{i=1}^{nK} \sum_{k'=1}^K \sum_{k=1}^K T_{k'k}^{-1} (y_{ik}^s - g_k(\boldsymbol{u}, \boldsymbol{x}_i^s))^2. \tag{5}$$

**Lemma 3.3.** *Let the notation be the same as before, the NTK kernel $\boldsymbol{G}(r)$ based on our proposed novel loss (5) is a block matrix with $K$ row partitions and $K$ column partitions. The block matrix in the $l$-th row and $m$-th column has the following form*

$$\boldsymbol{G}^{l,m}(r) = \begin{pmatrix} \mathcal{G}_{1,1}^{l,m}(r) & \mathcal{G}_{1,2}^{l,m}(r) & \dots & \mathcal{G}_{1,C}^{l,m}(r) & \mathcal{G}_{1,s}^{l,m}(r) \\ \mathcal{G}_{2,1}^{l,m}(r) & \mathcal{G}_{2,2}^{l,m}(r) & \dots & \mathcal{G}_{2,C}^{l,m}(r) & \mathcal{G}_{2,s}^{l,m}(r) \\ \vdots & \vdots & \ddots & \vdots & \vdots \\ \mathcal{G}_{C,1}^{l,m}(r) & \mathcal{G}_{C,2}^{l,m}(r) & \dots & \mathcal{G}_{C,C}^{l,m}(r) & \mathcal{G}_{C,s}^{l,m}(r) \\ \mathcal{G}_{s,1}^{l,m}(r) & \mathcal{G}_{s,2}^{l,m}(r) & \dots & \mathcal{G}_{s,C}^{l,m}(r) & \mathcal{G}_{s,s}^{l,m}(r) \end{pmatrix} \tag{6}$$

*for $l \in [K]$ and $m \in [K]$ where each sub-block matrix has the following form:*

$$\mathcal{G}_{c,j}^{l,m}(t) = \left\{ \sum_j Q_{jm}^{-1} \right\} \nabla_{\boldsymbol{u}} g_l(\boldsymbol{u}(t), \mathcal{D}^c) \nabla_{\boldsymbol{u}}^\top g_m(\boldsymbol{u}(t), \mathcal{D}^j), \ \forall j \in \{[C], s\}, c \in [C] \tag{7}$$

$$\mathcal{G}_{s,s}^{l,m}(t) = \left\{ \sum_k T_{km}^{-1} \right\} \nabla_{\boldsymbol{u}} g_l(\boldsymbol{u}(t), \mathcal{D}^s) \nabla_{\boldsymbol{u}}^\top g_m(\boldsymbol{u}(t), \mathcal{D}^s).$$

We show detailed derivation of Lemma 3.3. Note that as $J < K$, $\boldsymbol{Q} \in \mathbb{R}^{J \times K}$ is not inevitable. In practice, we compute its pseudo inverse instead. Although looking at $\boldsymbol{Q}$ alone, the plausible property of Theorem 3.1 may not hold. We claim that by optimizing $\widehat{R}_c(\boldsymbol{f})$ together with $\widehat{R}_s(\boldsymbol{f})$, we benefit from the gradient of labels in the desired label space, as shown in the expression of $\mathcal{G}_{c,s}^{i,j}(r)$ in Eq. (7).

Following Huang et al. (2021), we further conclude the convergence of the our proposed loss in (5).

**Theorem 3.4** (Convergence). *Let $M = \Omega(\lambda^{-4}(N + nK)^4 \log((N + nK)/\delta))$, we iid initialize $\boldsymbol{u}_m(0)$, $a_{km}$ as Definition 3.2. Let $\lambda = \lambda_{\min}(\boldsymbol{G}(0))$ denote the smallest eigenvalue of $\boldsymbol{G}(0)$. Let $\kappa$ denote the condition number of $\boldsymbol{G}(0)$. For $C$ clients, for any $\epsilon$, let*

$$R = O\left( \frac{C}{\lambda \eta_{\text{local}} \eta_{\text{agg}} t} \cdot \log(1/\epsilon) \right),$$

*$\eta_{\text{local}} = O\left(\lambda / \kappa t (N + nK)^2\right)$, and $\eta_{\text{agg}} = O(1)$, the above algorithm satisfies*

$$\mathcal{L}(\boldsymbol{u}(r)) \leq \left( 1 - \frac{\eta_{\text{agg}} \eta_{\text{local}} \lambda t}{2C} \right)^r \mathcal{L}(\boldsymbol{u}(0)).$$

*with probability at least $1 - \delta$.*

The details of the proof are deferred to Appendix B. Note that a smaller value of $\eta_{\text{agg}} \eta_{\text{local}} \lambda t / 2C$, *e.g.*, a smaller eigenvalue $\lambda$ and a larger number of clients, leads to slower convergence. It is clear that $\lambda$ depends on both $\boldsymbol{Q}$ and $\boldsymbol{T}$, therefore it is important to see its connection with $\boldsymbol{Q}$ and $\boldsymbol{T}$, we discuss the algorithm convergence under the following two special cases.

**Corollary 3.5** (Convergence under various label granularity differences). *Let $k_1, k_2, \dots, k_J$ be $J$ positive integers so that $K = \sum_{j=1}^J k_j$. If the transformation from the desired label space to the other label space is $\boldsymbol{Q} = diag(\mathbf{1}_{k_1}^\top, \mathbf{1}_{k_2}^\top, \dots, \mathbf{1}_{k_J}^\top) \in [0, 1]^{J \times K}$ (i.e., the classes in the desired label space are subclasses of classes in the other label space), then the smaller the value of $J$, the lower the convergence rate.*

---

[3]which can be easily extended to the label projection loss

*Proof Sketch:* Based on the form of $\boldsymbol{G}(r)$ in (7), it can be seen that (Li et al., 2021)

$$\lambda = \lambda_{\min}(\boldsymbol{G}(0)) \le \min_{k,c,s}\{\lambda_{\min}(\mathcal{G}_{c,c}^{k,k}(0)), \lambda_{\min}(\mathcal{G}_{s,s}^{k,k}(0))\} \le \lambda_{\min}(\mathcal{G}_{c,c}^{k,k}(0)) \lesssim \left\{\min_{j} k_j^{-1}\right\}\lambda_0$$

where $\lambda_0 = \lambda_{\min}(\nabla_{\boldsymbol{u}}\boldsymbol{g}(\boldsymbol{u}(0), \mathcal{D})\nabla_{\boldsymbol{u}}^{\top}\boldsymbol{g}(\boldsymbol{u}(0), \mathcal{D}))$. Since $\sum_j k_j = K$, the smaller the value of $J$, the smaller the value of $\lambda$ and hence the algorithm converges slower. See detailed proof in Appendix B.

**Corollary 3.6** (Convergence under different noise levels). *We assume noisy labels are corrupted via*

$$\boldsymbol{T} = (1 - K/(K-1)\xi)\,\mathbb{I}_K + \xi/(K-1)\mathbb{1}_K\mathbb{1}_K^{\top},$$

*where $T_{ij} = p(\widetilde{y}^s = Y_{\mathcal{F}}^j | y^s = Y_{\mathcal{F}}^i)$ with observed noisy label $\widetilde{y}^s$ and true $y^s$ is the true label, and $\xi = 1 - T_{ii} \in (0,1)$ for $i \in [K]$ denotes the noisy level. In this specified case, the convergence of* FedMT *with label projection on NTK is does not depend on the noise level $\xi$.*

*Proof Sketch:* By using matrix inversion lemma, we find $\boldsymbol{T}^{-1} = (K-1)/(K-1-K\xi)\mathbb{I}_K - \xi/(K-1-K\xi)\mathbb{1}_K\mathbb{1}_K^{\top}$. Hence $\sum_{k=1}^{K} T_{kk'} = 1$ for any $k' \in [K]$. See Appendix B for details.

## 4 EXPERIMENTS

In this section, we demonstrate the effectiveness of our proposed method, FedMT, when data are from different label spaces. Compared with other training strategies and prior art, FedMT consistently achieves better test accuracy in predicting labels in the desired label space with limited amount of data from this space as demonstrated on CIFAR100 (Krizhevsky et al., 2009) and a medical dataset.

### 4.1 BENCHMARK EXPERIMENTS SETUP

**Dataset and setting** We use the CIFAR100 (Krizhevsky et al., 2009) dataset to mimic our proposed problem setting. We assume one center (as server to coordinate FL training) has a small amount of data ($\le 5\%$ of total sample size) with sub-class annotations and other centers are labeled using super-class annotations. The sub-class space is viewed as our desired label space. Our objective is to train a classification model using FL to predict sub-class labels with all the centers simultaneously.

The CIFAR100 dataset consists of 50K images, associated with $K = 100$ sub-classes that could be further grouped into $J = 20$ super-classes. To study the effect of the number of training samples in desired label space, we randomly select $n$ observations from each of the 100 sub-classes on the server and the rest of the observations in the training set are split into $C = 10$ subsets completely at random, ensuring that each class is equally represented in each subset. Each subset corresponds to a dataset stored on one client and we use $N_c = 4000$. We use the super-class as our label for all clients. Experiment results with other values of $C$ and $N_c$ is given in Appendix D.

We use ResNet18 (He et al., 2016) as our classifier. We use SGD optimizer Ruder (2016) with a learning rate of $10^{-2}$, momentum 0.9, and weight decay $5 \times 10^{-4}$. The learning rate is divided by 5 at 20, 30, and 40 epochs. If not specified, our default setting for local update epochs is $E = 1$. Based on the superclass information in CIFAR100, we are able to formulate the transition matrix $\boldsymbol{Q}$ for all clients. Its form and more training details are available in Appendix C.1.

**Baselines** We compare the following approaches: 1) **Single**: only the data from the desired label space is used to train the classifier; 2) **FedMatch** (Jeong et al., 2021): we treat client samples as unlabeled data, perform pseudo-labeling on the unlabeled sets, and augment samples in supervised loss training. 3) **FedRep** (Collins et al., 2021): clients train a $J$-class classifier and the server trains a $K$-class classifier that differs in the last layer with different output dimension. These classifiers share the same backbone parameters using FedAvg, the performance is tested using the classifier on the server; 4) **FedTrans** (Chen et al., 2021b): all clients trains a $J$-class classifier using FedAvg which is later fine-tuned on server with a new $K$-class linear layer. We refer **FedMT (P)** as our proposed method with probability projection loss; and refer **FedMT (L)** as our proposed method with label projection loss.

### 4.2 BENCHMARK EXPERIMENTS RESULTS

**Comparison under different amount of data on server** We fix number of clients $C$ at 10, and set different number of samples per-class $n$ on the server in the range of $\{5, 10, 15, 20, 25\}$. The

Table 1: Comparison of the accuracy for 100-class classification using our methods and alternative methods on the CIFAR100 benchmark dataset with different per sub-class number $n$ images on the server. We report mean (sd) from three trial runs. The best method is highlighted in boldface.

| $n$ | Single | FedMatch | FedRep | FedTrans | *Ours*: FedMT (P & L) | |
|---|---|---|---|---|---|---|
| 5 | 8.56(0.33) | 13.20(0.14) | 16.89(0.74) | 21.31(1.27) | **21.68**(0.67) | 20.56(0.42) |
| 10 | 12.29(0.32) | 18.02(0.25) | 21.90(1.27) | 24.53(1.40) | **26.44**(0.34) | 23.89(0.13) |
| 15 | 14.34(0.15) | 23.44(0.08) | 23.39(0.67) | 26.21(0.95) | **28.25**(0.34) | 25.74(0.59) |
| 20 | 16.23(0.21) | 27.70(0.09) | 24.70(0.55) | 27.49(0.85) | **29.75**(0.66) | 27.43(0.35) |
| 25 | 17.73(0.20) | 30.71(0.59) | 26.53(0.29) | 28.16(1.30) | 30.63(0.06) | 28.53(0.52) |

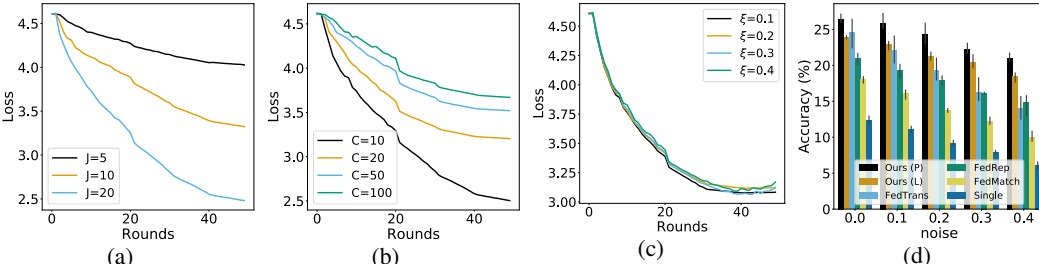

Figure 2: Ablation studies on CIFAR100. (a) Effects of the number of classes $J$ on convergence. (b) Effects of the number of clients on convergence. (c) Effects of noise level $\xi$ on convergence. (d) Study of performance accuracy with different noise levels on the server. All the convergence results are shown on the first 50 rounds using FedMT (L) for better visualization.

advantage of FL is to enable access to more data to improve model performance. As shown in Tab. 1, when $n$ increases (supervision information increases), the accuracy of all approaches improves. Both FedMT (P & L) significantly outperform alternative methods when supervision is limited $n < 25$. We achieve similar but much more stable performance compared with FedMatch when $n = 25$. FedMatch adaptively augments the training data with labels in the desired label space using heuristically derived confident pseudo labels. Future work could combine FedMT with similar approaches used in FedMatch to evaluate heuristic results in a theoretically provable framework.

**Analysis on convergence** To validate our developed theoretical results in Theorem 3.4, we investigate the convergence rate of loss (2) of FedMT with label projection. In Fig. 2(a), we explore loss as a function of the number of classes $J \in \{5, 10, 20\}$ with $C = 10$ under noise-free setting, and the convergence is consistent with Corollary 3.5, smaller $J$ leads to lower convergence. The loss curves in Fig. 2(b) show FedMT convergence was achieved faster when training with more clients for a fixed total number of samples. This result is consistent with the theoretical result of Theorem 3.4. As shown in Fig. 2(c), under the default setting, the convergence rates are similar at different noise levels. The results demonstrate our method is reasonably stable to noise, which is in line with Corollary 3.6.

**Effect of varying noise levels** Our method has the flexibility to adapt to the case where the server data is noisy. We use the accuracy of $K$-class prediction in the held out test set to compare our method with other approaches in CIFAR100, the results are reported in Fig. 2(d). Given the mislabeling rate in many real applications (*e.g.,* healthcare) could be as high as $40\%$ (Lettieri et al., 2005), we compare FedMT with other learning approaches, by varying the noise level $\xi$ on the server from $\{0, 0.1, 0.2, 0.3, 0.4\}$ and set $n = 10$ on the server.[4] From the results, our proposed FedMT method significantly outperforms all alternative methods both even when the noise level $\xi$ is up to $0.4$. Second, when increasing noise levels, FedMT shows more stable performance compared to baselines. Our method also shows stable performance if we have more than 1 client has observations from desired label space and the results is given in Appendix D.5.

### 4.3 TREMOR SEVERITY PREDICTION OF PARKINSON'S DISEASE

**Dataset** Our method is not limited to sub- and super-classes as in CIFAR100, to illustrate the effectiveness of our method under the general overlapping class case, we evaluate FedMT for the

---

[4]We assume the labels are sampled from known, systemic, and instance-independent noisy transition matrix $T$, whose diagonal values are $1 - \xi$. Our method can be extended to estimate $T$ using anchor point-based methods (Scott, 2015; Liu & Tao, 2015) but discussing the estimation method is not in the scope of this work.

Table 2: Comparison of the accuracy for 5-class classification using our methods and alternative methods on the sEMG dataset with various $n$ and $C = 50$ under noise-free setting. We report mean (sd) from three trial runs. The best method is highlighted in boldface.

| $n$ | Single | FedMatch | FedRep | FedTrans | *Ours*: `FedMT` (P & L) | |
|---|---|---|---|---|---|---|
| 1 | 27.30(0.88) | 28.20(0.67) | 39.34(1.12) | 42.31(2.12) | **66.89**(0.91) | 63.67(0.28) |
| 2 | 30.83(1.30) | 31.90(1.45) | 40.49(1.16) | 46.45(0.63) | **67.00**(0.38) | 64.22(0.28) |
| 3 | 34.66(1.09) | 32.90(1.74) | 41.29(1.16) | 51.25(2.57) | **67.34**(0.57) | 64.33(0.48) |
| 4 | 39.11(1.48) | 33.53(2.12) | 43.24(0.68) | 51.81(1.99) | **67.67**(0.64) | 64.44(0.35) |
| 5 | 42.57(0.78) | 37.20(2.09) | 44.24(0.74) | 51.83(2.14) | **67.22**(0.30) | 64.44(0.55) |

prediction of tremor severity for patients with Parkinson's disease (PD). Tremor is a typical movement disorder occurring on the limbs of patients with PD (Qin et al., 2020). Surface electromyography (sEMG) is widely used for movement disorder assessment by noninvasively recording electrical signals on the skin. The patterns of sEMG signals can be used to turn PD diagnosis into different subtypes based on the diagnostic values. We use the synthetic sEMG dataset in Qin et al. (2020). In our experiment, we consider $K = 5$ on the server and $J = 3$ on the client, according to MDS-UPDRS19 Goetz et al. (2008).[5] In this dataset, the classes in two label spaces has overlap. The form of $Q$ is given in Appendix C.2.

**Setup** We randomly sample 9000 observations from the sEMG dataset as client training data. We hold out 1000 observations for testing. We then split the training set to $C = 50$ clients completely at random, but ensuring that samples are balanced within each class in both label spaces. Each client, therefore, has roughly 60 observations from one of the three classes. In addition to the training set on clients, we also have $n$ observations per class on the server. Similar to the benchmark experiment, we vary $n$ on the server. Following Qin et al. (2019), we use 12 summary statistics of sEMG as our features: mean absolute value (MAV), mean square value (MSV), root mean square (RMS), variance (VAR), standard deviation (STD), waveform length (WL), Willison amplitude (WAMP), log detector (LOG), slope sign change (SSC), zero crossing (ZC), mean spectral frequency (MSF) and median frequency (MF). The mathematical definition of these features is given in Chowdhury et al. (2013). We then use a single hidden layer multiple layer perceptron (MLP) with 128 hidden units and ReLU activation function as our backbone. We use SGD optimizer with learning rate as $10^{-3}$, and set batch size as 16. Local update frequency is 1 epoch. We train the model for 100 communication rounds. More training details are available in Appendix C.2. We use the same performance measure and the same alternative approaches as those described in Section 4.1 for comparison.

**Results** Overall, our quantitative results in Tab. 2 show that `FedMT` with probability projection performs better than probability projection on the sEMG dataset and consistently outperforms alternative learning approaches. `FedMT` improves mean accuracy by a non-negligible margin, demonstrating a synergistic collaborative effect with clients' data from the other label space. The superiority of `FedMT` in this demanding medical data setting further demonstrates the efficacy and robustness of our algorithm. To test of the performance of `FedMT` at different degree of overlapping, we also conduct experiments when $K = 10$ with various noisy levels, the results is given in Appendix E. Our method also outperforms other methods in this case.

## 5 CONCLUSION

In this work, we propose a new FL framework, `FedMT`, to address an important yet under-explored mixed-type label setting. Theoretically, we provide the convergence guarantee of `FedMT` with the extension of NTK. Through extensive experiments on a benchmark and a medical dataset, we demonstrate `FedMT` can outperform alternative methods. We also emphasize that since `FedMT` makes minor modifications to local predictions or labels, it has much more flexibility to integrate with other FL strategies beyond FedAvg. The performance of our proposed method can be further improved by combining our provable method with other heuristic-based weakly supervised learning approaches. As a plug-and-play method, `FedMT` can be applied to non-IID setting by combining with advanced FL schemes, including Li et al. (2020b; 2021); Karimireddy et al. (2020).

---

[5]The labeling strategies are detailed in Appendix C.2

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
