# OpenReview forum: "FedMT: Federated Learning with Mixed-type Labels"
_ICLR.cc/2023/Conference — Submitted to ICLR 2023_

### Official Review · Reviewer_xV7p · 2022-10-13

**Confidence:** 3
**Clarity, Quality, Novelty And Reproducibility:** This paper is easy to read, and the n…
**Correctness:** 3
**Technical Novelty And Significance:** 3
**Empirical Novelty And Significance:** 3
**Recommendation:** 8

**Strength And Weaknesses:**

Strength:
1. This paper is the first work to investigate FL’s theory on neural network classification under mixed-type labeled data.

2. This paper is easy to read.

3. The authors propose a theory-guided and model-agnostic approach that can use the underlying correspondence between those label spaces and can be combined with various FL methods.

4. The proposed method achieved relatively good performance than the baseline methods.

Weaknesses:
1. The authors implement the experiments with the CIFAR100 dataset and ResNet18. I am curious about the experiments with large datasets, such as ImageNet, or lightweight architectures, like MobileNetV2.

2. The authors propose two kinds of projections for servers and clients, as in Eq. (3) and Eq. (4). I am curious about these two projections' effects on the proposed algorithm.


**Summary Of The Paper:**

In this paper, the authors propose a method that implements the federated learning(FL) with mixed-type labels. In detail, the authors propose a theory-guided and model-agnostic approach that can make use of the underlying correspondence between different label spaces and can be easily combined with various FL methods. The proposed method achieves better results than the baseline methods.

**Summary Of The Review:**

In this paper, the authors propose a method that implements the federated learning(FL) with mixed-type labels. In detail, the authors propose a theory-guided and model-agnostic approach that can make use of the underlying correspondence between different label spaces and can be easily combined with various FL methods. The proposed method achieves better results than the baseline methods. The novelty is relatively good, and the proposed method achieved a relatively good performance than the baseline methods.

---

### Official Review · Reviewer_B327 · 2022-10-22

**Confidence:** 3
**Correctness:** 3
**Technical Novelty And Significance:** 3
**Empirical Novelty And Significance:** 3
**Recommendation:** 8

**Clarity, Quality, Novelty And Reproducibility:**

This paper targets an interesting scenario in federated learning where each center has different labeling criterion, and the proposed algorithm is also simple and efficient, which could benefit the federated learning community. In terms of reproducibility, the authors provide clear description of experiment setup and the details are also included in the supplemental material.

**Strength And Weaknesses:**

Pros:

The paper is well-written and well-motivated. The discussed topic is challenging and important to the federated learning in real scenarios. The proposed projections are easy to follow and seem natural for the mixed-type labels. The authors provide theoretical analysis of the convergence of proposed algorithm. The experiments are extensive. The evaluation is conducted on various datasets and the proposed algorithm is compared to various SOTA baselines. The results are promising and convincing.


Cons:

My major concern lies in the advantage of two proposed projections. The authors introduce two projections align the prediction of different types of labels to the true label space, which are applied to probability and label respectively. Although the theorical analysis focuses on the label projection and shows better convergence, the results in Table 1 and Table 2 show the superiority of probability projections over label projections. It is difficult to see the necessity of proposed label projections.


**Summary Of The Paper:**

The authors propose to tackle the potential mixed-type label problem in federated learning. Specifically, two types of projections are introduced to both server and clients to achieve the alignment, which can be also generalized to noisy label scenario. The authors also provide theoretical convergence analysis. In the experiments, the authors conduct evaluation on CIFAR-100 and sEMG datasets, the proposed projections can significantly improve the performance compared to other SOTA federated learning baselines.

**Summary Of The Review:**

Overall, I think this paper is interesting and solid. It would be better if the authors can provide more discussion of two types of projections and demonstrate the scenario where probability/label projection shows superiority.

---

### Official Review · Reviewer_23rr · 2022-10-25

**Confidence:** 4
**Correctness:** 3
**Technical Novelty And Significance:** 3
**Empirical Novelty And Significance:** 3
**Recommendation:** 6

**Clarity, Quality, Novelty And Reproducibility:**

The paper is clearly written and easy to follow, the proposed approach is clearly motivated and supported by theoretical findings. Novelty is somewhat limited considering the straightforward extension of the work in (Van Rooyen & Williamson, 2017; Patrini et al., 2017) to the federated learning setting, however, it is worth noting that the authors leverage results from NTK to better understand the convergence of the proposed approach, even considering the effects of Q and T on convergence.

**Strength And Weaknesses:**

The proposed approach is well motivated, justified from a practical perspective and supported by theoretical results. The experiments demonstrate both the performance of the model relative to competing approaches and support the theoretical findings. The main weaknesses are: i) the need for known and fixed Q and T matrices, ii) the assumption that only the specialized server suffers from label noise, iii) that though the proposed approach is proposed as a federated learning approach, the fact that only two sets of labels are considered (one containing the other) reduces the federated aspect to the model to simply applying FedAvg with the corrupted label learning of (Van Rooyen & Williamson, 2017; Patrini et al., 2017), which can be seen by comparing Algorithms 1 and 2. Note though that the latter could be perceived as a strength of the proposed approach except that it clearly separates the members of the federation as i) those with the larger set of labels which may be affected by noise, and ii) those with a smaller but common set of labels that are not only cheaper to obtain but are noise free. These two assumptions greatly limit the applicability of the proposed approach in general federated learning settings.

The authors make the assumption that there are only two label spaces, however, such an assumption lacks a practical justification, more so considering the original motivation of the approach (from the paper: "standards for disease diagnosis may be different across clinical centers due to varying levels of expertise or technology available at different sites"). Though, the authors claim that extension is easy, it is not clear why being easy, is not explored.

The structure of the proposed framework raises several questions that are not addressed in the paper:
- why there is a need for a server that both handles the aggregation but also contributes data. This is not standard in FL, arguably unnecessary, and may result in privacy issues or at the very least power imbalance issues (the specialized server has access to the model, and updates from all the clients while at the same time having a local model and access to data).
- why only assume that the specialized server has noisy labels?
- from the text it is not clear under which assumptions does Theorem 3.1 holds and how those relate to the practical setting considered by the authors.

For a benchmark experiment, it is justifiable that T and Q are not only known but easy to obtain, however, in practice, estimating T is in general difficult and Q is complicated for non-nested cases.

The paper specifies that the PD data from Qin et al. (2020) is synthetic, however, other places in the text and the supplementary material suggest that the data is real. Can the authors please clarify? The reasoning for the question is to better understand how realistic are the assumptions about Q and T in relation to whether the data, endpoints, noise or sub-classes (with overlap) were artificially crafted.


**Summary Of The Paper:**

The authors address the problem of federated learning in the scenario where clients have differently defined label sets (assuming a super set of labels and a subset with and without partial overlap), but without requiring cross-client data relabeling. Further they consider also the setting where only a small portion of the data (in the specialized server) is labeled and labels may be noisy, They also provide theoretical results concerning the convergence of the proposed approach.

**Summary Of The Review:**

The proposed approach constitutes a first step in the development of federated learning systems with nested or partially overlapping endpoints, while allowing for clients/servers with small sample size and label noise. However, the assumptions (known and fixed Q and T, one common subset of labels for all clients, noise labels only in the server, etc.) and experiments underscore the limitations of the proposed approach. Specifically, it narrows down the applicability of the proposed approach to very specific and arguably rare practical situations.

---

### Official Review · Reviewer_N8Hr · 2022-10-28

**Confidence:** 3
**Correctness:** 2
**Technical Novelty And Significance:** 2
**Empirical Novelty And Significance:** Not applicable
**Recommendation:** 3

**Clarity, Quality, Novelty And Reproducibility:**

Clarity: as mentioned above, there are several issues, e.g., design of T,Q, convergence results, which are clearly discussed.

Quality: the theoretical part may contain some errors. SGD generally cannot achieve the linear convergence rate, even in the strongly convex case. The number of inner loop $t$ times the step size is appeared in the convergence result, meaning that if the server and the nodes do not communicate frequently, then, the local nodes are required to choose small step sizes (which is inversely proportional to $t$). Why does not use a large step size? This claim is not strong.

Novelty: I believe this setting is new. The error correction idea is actually borrowed from the centralized one directly. If the major differences between these settings can be further addressed, the novelty of the designed method should be good.

Reproducibility: the code is available at an anonymous link.

**Strength And Weaknesses:**

The strength of this work is mainly about the consideration of a new setting in the FL system. The mismatch among the center do exist in practice. To the best of my knowledge, this would be the first work addressing this issue. The proposed algorithm seems simple and easily implementable. The numerical results are encouraging.

The main weaknesses of this work are as follows:

1) it is quite confusing about the projection matrices. It seems that this idea is borrowed from the other (centralized one). There are many issues missing, e.g., how to choose these matrices so that they are invertible? it seems that they are given as prior. Then, why do claim the agonistic?

2) The convergence results are built upon the NTK regime. Why is this regime considered? Convergence on the first-order stationarity for nonconvex loss function is widely used and applicable for general problem beyond neural nets.

3) It is claimed that even SGD used in this work can achieve the linear convergence rate and a step size of independent on iteration is used. Then, how about the gradient descent case? How does stochastic gradient estimate error play into the convergence result?



**Summary Of The Paper:**

This paper mainly focuses on a practical scenario of FL where the labels might be marked by different criteria over different centers/nodes. A new learning strategy is proposed for dealing with the mismatches of the labels, and it can be deployed with existing FL methods. Theoretical analysis shows that the proposed FedMT is able to find the global optimal solution for this class of problems at a linear convergence rate. Multiple numerical results further verify the intuition of this method with the experiments on real datasets.

**Summary Of The Review:**

In summary, the work indeed brings the mismatch issue in the FL setting, but a thorough discussion on FedMT should be carefully studied, including projection matrix design, convergence analysis, communication efficiency, etc.

---

> ### Author Response · Authors · 2022-11-05
> **Thanks for considering our typo amending**
>
> We really appreciate your careful review. In our analysis of the NTK regime, we fully agree with the gradient descent assumption. Actually, we meant gradient descent (gd) rather than (sgd) in our NTK theoretical analysis of Theorem 3.4. The typo of $\eta$ was caused by accident in the macro definition of the math commands. We sincerely apologize for the confusion.
>
> As you can see in our related work (e.g., “training an infinitely wide neural network with gradient descent”) and our theoretical proof in Appendix B (e.g., “Let $\eta$ be the learning rate of the gradient descent algorithm” in B.1 and “the evolution of parameters $u$ and the output $g$ via continuous-time gradient descent satisfies...” in B.2), we consistently stated that we considered “gradient descent” in our theoretical analysis under NTK setting.
>
> In our revision, we have corrected the notation $\eta_{sgd}$ to $\eta_{local}$ and highlighted that the NTK analysis was based on gradient descent in Section 3.4. We also noted the readers that in the real implementation, we use SGD for local training for efficiency following the existing work on NTK or motivated by NTK [1,2]. We'd be grateful if the reviewer could pardon us for this typo and consider our correction.
>
> Thanks for your other valuable comments. We will address the remaining comments carefully soon.
>
> [1] A Neural Tangent Kernel-based Framework for Federated Learning Analysis (ICML21)
>
> [2] Picking Winning Tickets Before Training by Preserving Gradient Flow (ICML20)

---

> ### Author Response · Authors · 2022-11-24
> **Appreciate your feedback to our response**
>
> Dear reviewer N8Hr,
>
> Thank you for the detailed review. We hope that we have clarified your questions and provided you with sufficient evidence to raise your score. Please see our responses below. We are delighted to answer your remaining concerns. We would appreciate your positive feedback if you are satisfied with our responses. We are grateful for your time and consideration.
>
> Best,
>
> FedMT authors

---

> > ### Comment · Reviewer_N8Hr · 2022-11-25
> > **There are still some concerns**
> >
> > Great. Thanks for your time in addressing my comments. First, regarding your response, I am totally fine that the SGD that appeared in the algorithm and theory is a typo. Also, I agree that the proposed method is novel in the sense that it considers a practical learning issue (mixed-type labels) in the FL setting.
> >
> > However, there are still many issues in this paper, which are mainly about the technical concerns and quality of the presentation.
> >
> > 1, communication efficiency means how many $t$ that the algorithm can afford so that the algorithm can still find the stationary or optimal points. What is the theoretical bound of $t$ here?
> >
> > 2, The authors argued that ‘FedMT is built on FedAvg’. I agree with that. But in this setting, due to the mixed-type labels, the network is heterogeneous. However, it is well known that FedAvg cannot find the stationary or points even the problem is strongly convex (please see Theorem 4 in [R]) when a constant step size is used for GD. I think the convergence analysis is built on the homogeneity of the network (data distributions). A detailed discussion is needed.
> >
> > [R] Li et al, On the convergence of FedAvg on Non-iid data, ICLR, 2020.
> >
> > 3, NTK: using NTK is only reasonable for over-parameterized neural networks. Is the neural net used in this work overparameterized? Also, it is very clear that in Theorem 3.1 T has played the role in the loss correction, so the \beta-smooth parameter will be affected by the projection for sure, which is consistent with G used in NTK.
> >
> > 4,  a fixed large local update step means $t=1$. The current theorem covers this case. My concern was when $t=1$, a properly chosen step size, say $\eta$, can converge. However, when $t>1$, say $t=10$, we have to choose a step size with the order of $\eta/10$, right? Noe that the convergence rate is the same. If we can choose $\eta$ with one step update, why should we artificially select a small step and run for multiple steps?  Is it a waste of computational time or resources?
> >
> >
> > AC had further initialized a detailed discussion. To my reading, there are still many issues regarding the presentation as well as the clearness of the technical derivation in this work.
> >
> > During the update of $u_{m,c}$, a partial derivative is used due to the non-smoothness of the activation function, but in the algorithm, the vanilla gradient is used, which is not consistent with the update equation. Please explain.
> >
> > In the update equation of $u_m$, why does the algorithm perform gradient descent? Is there a need to increase the objective function? Again, it is still NOT consistent with the algorithm described.
> >
> > Why does $\sum^K_{k=1}T_{k’k}=1$ lead to linear convergence?
> >
> > Theorem 3.4 holds only a special $T$, right? In the proof, a special $T$ was selected. Why does not mention this special in the statement of the theorem?
> >
> > In the theorem, the authors show the convergence for gradient descent while in the numerical results, SGD is used, leading to a gap between the theory and experiments. So, how possible to claim consistent between the theory and algorithm on page 8? Also, why does SGD still have a momentum term? Is it accelerated SGD?
> >
> > Figure 2, it seems that the tested algorithm has not achieved a stationary point. How about other algorithms?
> >
> > In the discussion of the experiment results, it is mentioned that the noise level affects the performance, while in the theory it said no. It is also observed in Tables 9 and 11. Is there a contradiction?
> >
> > In the second experiment, why is only SGD used rather than SGD with momentum?
> >
> > Please check ‘Methodologically, we …solution Theoretically, we … levels; Empirically, we … datasets.’ I double the punctuation mark or capitalized characters were used correctly.
> >
> > on $c$-th client or on the $c$-th client?
> >
> > Page 4, $j$th class or the $j$th class?
> >
> > The sentence “…, that is the label is…” is not well written.
> >
> > In the algorithm, the model update is represented by $f$, while in the update of the model in equations (equations numbers are missing) is represented by $u$. What are the reasons for using different characters to represent the model update and model update weight? Are these differences critical? It seems that the equations are exactly the same.
> >
> > Page 5, $c$th client or the $c$th clinet? $r$th global round or the $r$th global round?
> >
> > What does mean of ‘$Q$ is not inevitable.’ On page 6? Is it ‘invertible”?
> >
> > Page 16, the sentence ‘…is also consisted of block…’ is not well written. Please check.
> >
> > The notations of $Q$ and $Q^{-1}$ are not consistent on page 17.
> >
> > Last but not least, why is the code link deleted in this version? This will affect the reproducibility of this work.

---

> > > ### Author Response · Authors · 2022-11-26
> > > **Responses to Reviewer N8Hr's 2nd Round of Feedback**
> > >
> > > Dear reviewer, we highly appreciate the reviewer for the careful review. As it already passed stage 1, we are not able to fix the typos and update our revision. We will polish the writing following your suggestion in our final version. We hope you like our problem setting and the method we propose to tackle the problem that is essential under FL. Thank you again for your constructive feedback, we appreciate your valuable input and your time. We are happy to address any additional concerns that you may have, and we really appreciate if you could reevaluate our paper based on the discussion. Please see our detailed point-to-point responses below to address the reviewer's concerns raised in the 2nd round of feedback: Part 1 - to your question 1 - 3; Part 2 - to your question 4 and its following up questions; and Part 3 - other issues.

---

> > > > ### Author Response · Authors · 2022-11-26
> > > > **Part I – response to question 1 to 3**
> > > >
> > > > > communication efficiency means how many $t$ that the algorithm can afford so that the algorithm can still find the stationary or optimal points. What is the theoretical bound of $t$ here?
> > > >
> > > > Thanks for the great question and we agree that $t$ affect communication efficiency. Following the existing main stream work on FL convergence analysis [1-3] with a fixed total update epochs $t$ (note [1,2] denote it as K and [3] denotes it as E), we showed how to set proper learning rate for a given $t$ to ensure convergence. Nevertheless, if given $\eta_{agg}$ and $\eta_{local}$, it won’t be hard to see $t\geq\max\(\left({O(\lambda/\kappa(N+nK)^2\eta_{\text{local}}), O(\lambda/\kappa(N+nK)^2\eta_{\text{local}} \eta_{\text{global}})\}\right)$ (amended) following the proof in B.1 of [1].
> > > >
> > > > > The authors argued that ‘FedMT is built on FedAvg’. I agree with that. But in this setting, due to the mixed-type labels, the network is heterogeneous. However, it is well known that FedAvg cannot find the stationary or points even the problem is strongly convex (please see Theorem 4 in [R]) when a constant step size is used for GD. I think the convergence analysis is built on the homogeneity of the network (data distributions). A detailed discussion is needed.
> > > >
> > > >
> > > > Thanks for the question. We'd like to address that [R]'s Theorem 4 is NOT applicable in our theoretical analysis settings and our convergence analysis does not have to reply on homogeneity of the network. In fact,
> > > >
> > > > - [R] and our convergence analysis are based on very different settings and assumptions, thus the conclusion of [R] is not applicable in our setting. can Specifically, [R] follows classical (non-)convex optimization with the first-order stationary method with certain assumptions and we agree typically it requires decreasing learning rate schedule by such an approach. However, our theory analysis is on the over-parameterized regime for neural nets, where literature has shown that the model can achieve zero training error, namely reaching optimum [1,4,6]. Also, in the over-parameterized regime, [1,4] showed convergence with a constant learning rate.
> > > >
> > > > - The analysis of NTK convergence in the over-parameterized regime does not need to make explicit assumptions on data distribution [1,4]. Instead, the data sampled from certain distribution(s) will directly be reflected in the kernel matrix, whose smallest eigenvalue affects the convergence. [5] has already justified the convergence results in a unified form for both iid and non-iid data. Lastly, mixed-type data does not necessarily mean heterogeneous distribution wrt desired label space.
> > > >
> > > >
> > > > > NTK: using NTK is only reasonable for over-parameterized neural networks. Is the neural net used in this work overparameterized? Also, it is very clear that in Theorem 3.1 T has played the role in the loss correction, so the \beta-smooth parameter will be affected by the projection for sure, which is consistent with G used in NTK.
> > > >
> > > > We agree that there is a gap between theory and practice. But we don’t think such a gap will affect the contribution of our work that proposes a novel FL framework while providing theoretical implications. Furthermore, as indicated by our experiments, the theoretical results are still validated in more practical settings. As we listed in our original response, there are many published works that perform theoretical analysis under overparameterized neural networks but perform experiments under limited-width neural networks using SGD [1,7]. Alternative convergence analysis tools still face the issue that there exists a gap between theory and practice, for example the assumptions on the loss functions might not be realistic.
> > > >
> > > > References
> > > >
> > > >
> > > > [1] FL-NTK: A Neural Tangent Kernel-based Framework for Federated Learning Convergence Analysis. (ICML 2021)
> > > >
> > > > [2] SCAFFOLD: Stochastic Controlled Averaging for Federated Learning. (ICML 2020)
> > > >
> > > > [3] (or [R] ) On the convergence of FedAvg on Non-iid data, (ICLR 2020).
> > > >
> > > > [4] On Exact Computation with an Infinitely Wide Neural Net. (NeurIPS 2019)
> > > >
> > > > [5] A Convergence Theory for Deep Learning via Over-Parameterization. (ICML 2019)
> > > >
> > > > [6] The Impact of Neural Network Overparameterization on Gradient Confusion and Stochastic Gradient Descent. (ICML 2020)
> > > >
> > > > [7] Picking Winning Tickets Before Training by Preserving Gradient Flow (ICML20)

---

> > > > > ### Author Response · Authors · 2022-11-26
> > > > > **Part II – Clarification on the theoretical analysis**
> > > > >
> > > > > > a fixed large local update step means $t=1$... If we can choose $\eta$ with one step update, why should we artificially select a small step and run for multiple steps? Is it a waste of computational time or resources?
> > > > >
> > > > > Thanks for the questions. Does $t>1$ make no sense? NO! This is because using $t=1$ and a large learning rate, it may not meet the requirement of bounding the global movement model weights in the NTK regime as we stated in our original response. Let us think step by step. First, follow [1], we have
> > > > > $||y_c^{(k+1)}(t) - y_c^{(k)}(t)||_2^2 \leq \eta  n^2 || y_c^{(k)}(t) -y_c||_2^2$
> > > > >
> > > > > where $\eta$ is the local step size. It is obvious to see that a large $\eta_{\text{local}}$ will lead to a large bound. Second, the change of global model weights will also have a large bound based on Lemma B.10 of [1]. As a result, the global model movement will have a larger upper bound based on Lemmar 6.1 or Lemma B.12 of [1]. It is a common standard in NTK analysis that model weights in the NTK regime stay stable as we stated in our original response, which is also shown in [1,4] that the bounds of model weights movement should be sufficiently small. Also, we mentioned in our original response that the relationship between $t$ and $\eta$ is also consistent with the results presented using other convergence analysis tools.
> > > > >
> > > > > p.s. FYI, we used different notations. [1] use $t$ for global iteration, and $K$ for local updates and we used $\tau$ and $t$ respectively.
> > > > >
> > > > > >In the algorithm, the model update is represented by $f$, while in the update of the model in equations (equations numbers are missing) is represented by $u$. What are the reasons for using different characters to represent the model update and model update weight? Are these differences critical? It seems that the equations are exactly the same.
> > > > >
> > > > > We would like to point out that there was NO conflict. As you can see from the equation under our theoretical NTK setup, $f$ is an explicit function of parameter $u$. Under NNs, the update on $f$ by default is the update on its parameter $u$. In the algorithm description, we omitted $u$ for simplicity and for formatting our manuscript in a nice way. It seems the this did not affect other reviewers’ understanding. If reviewer N8Hr strongly insists putting $u$ back to the algorithm box, we are delighted to address it in our final version.
> > > > >
> > > > >
> > > > >
> > > > > > During the update of $u_{m,c}$, a partial derivative is used due to the non-smoothness of the activation function, but in the algorithm, the vanilla gradient is used, which is not consistent with the update equation. Please explain.
> > > > >
> > > > > We would like to clarify they are **consistent**. The question is closely related to the previous question we answered above. The gradient of of an objective function with respect to all its parameters is the concatenation of all the partial derivatives of the objective function with respect to different parameters. For $u_{m,c}$, we specifically write out its form and use a general notation for $f$ for preciseness. We are happy to use a consistent notation in our final version if the reviewer insists.
> > > > >
> > > > >
> > > > > > In the update equation of $u_{m}$, why does the algorithm perform gradient descent? Is there a need to increase the objective function? Again, it is still NOT consistent with the algorithm described.
> > > > >
> > > > > We assume that you meant “gradient ascent” not “gradient descent”. $u_m$ is indeed performing gradient descent. Here is our clarification.
> > > > >
> > > > > Thanks for pointing it out. The aggregation step (line 9 of algorithm 1 and line 14of algorithm 2) should have a positive sign before $\eta_{agg}$, that is $f \gets f + \eta_{\text{agg}} \sum_{c} (f^{c} -f).$ This is because in the gradient descent $w^{t+1} = w^{t} - \nabla f$, namely $\nabla f = w^{t} -w^{t+1}$. Then $f^{c}-f$ can be viewed as the negative 'gradient' since $f$ is the weight from last step in the aggregation step. The same, $\sum_{c}\Delta u_{m,c}(r) = u_{m,c}(t,r) - u_{m,c}(0,r)$ (after updating - before updating) can be viewed the negative 'gradient' direction and it is update equation of gradient descent is correct.
> > > > >
> > > > >
> > > > > > Why does $\sum_{k=1}^{K} T_{k’k}=1$ lead to linear convergence?
> > > > >
> > > > > Do you refer to the last line of page 16? As we show in Theorem 3.4, the linear convergence rate of the algorithm under the NTK setting holds for a general transition matrix $T$. The special case of $\sum_{k=1}^{K} T_{k’k}^{-1}=1$ that you referred here is a corollary of the theorem.
> > > > >
> > > > > > Theorem 3.4 holds only a special $T$, right? In the proof, a special $T$ was selected. Why does not mention this special in the statement of the theorem?
> > > > >
> > > > > We want to address that Theorem 3.4 and its proof do NOT depend on the specific form of $T$. The theorem holds for a general $T$ and the proof under the general $T$ is given in Appendix B.2. We discuss the special form $T$ in the corollary to show that under our benchmark experiment setting, the value of $J$ does not affect the rate of convergence.

---

> > > > > > ### Author Response · Authors · 2022-11-26
> > > > > > **Part III - Other Issues**
> > > > > >
> > > > > > > Figure 2, it seems that the tested algorithm has not achieved a stationary point. How about other algorithms?
> > > > > >
> > > > > > We want to clarity the figure 2 you refer to did not include all the training iterations. We provided the experimental details in Appendix D saying that we trained for 100 rounds. The curves for 100 rounds are also available in Figure 3 where losses are all converged. We presented the curve for the first 50 rounds to highlight the differences and similarities. We are delighted to highlight the details in our final version.
> > > > > >
> > > > > > > In the discussion of the experiment results, it is mentioned that the noise level affects the performance, while in the theory it said no. It is also observed in Tables 9 and 11. Is there a contradiction?
> > > > > >
> > > > > > No, it is not a contradiction. The theory, table 9, and table 11 are different things.
> > > > > >
> > > > > > The theory is on the convergence analysis of the algorithm. By perform of a method, we usually refer to the test accuracy and we did not claim “noise level does not affect the performance” in our theory. Instead, we carefully stated that **the convergence of** FedMT with **label projection** on NTK.” The result in corollary 3.6 that the noise level does not affect the convergence rate is validated empirically in Figure 2 (c).
> > > > > >
> > > > > > Table 9 reports the classification accuracy of FedMT with probability projection when the server data is contaminated with noise. We apologized that we might not state it clearly in the Table 9 caption but we will add it to our final version.
> > > > > >
> > > > > > Table 11 reports the result of the sensitivity analysis of the robustness of $T$. That is when the true $T$ is disturbed by a noise level, how does our proposed method perform?
> > > > > >
> > > > > > In summary, the reviewer pointed to three different things and there is no contradiction. We are happy to change the notation and add an additional explanation if this confused the reviewer.
> > > > > >
> > > > > >
> > > > > >
> > > > > >
> > > > > >
> > > > > > > In the second experiment, why is only SGD used rather than SGD with momentum?
> > > > > >
> > > > > > As we stated in Appendix B and Appendix C.1 and C.2, we have used SGD with momentum on both the benchmark experiment and the medical experiment.
> > > > > >
> > > > > >
> > > > > >
> > > > > > > why is the code link deleted in this version?
> > > > > >
> > > > > > Oops, the link was missing unintentionally. We are delighted to share our code which is available via this anonymous link: shorturl.at/bdgjw. It is the same as the original submission. Also, no file is changed as evidenced by the time stamps.
> > > > > >
> > > > > > > The sentence “…, that is the label is…” is not well written.
> > > > > >
> > > > > > Sorry, we meant “Another advantage of our method is that it has the flexibility to extend to the case where there are noisy observations. For noisy observation, we refer to the case where the actual label of an observation is mislabeled as a wrong one in the same label space”

---

> > > > > > ### Comment · Reviewer_N8Hr · 2022-11-26
> > > > > > **need to see the proof.**
> > > > > >
> > > > > > I am confused about your response " using $t=1$... it may not meet the requirement of bounding the global movement ...". Theorem 3.4 should be true all $t$, right? (now you mentioned that there was an error because $t$ should have an upper bound, right?, please correct me if I am wrong. I suppose $t$ should not be too large, e.g., in the order of $\epsilon$, otherwise, it will affect the final rate.) So, when $t=1$, the algorithm should work, and my comments are applied.
> > > > > >
> > > > > > Also, I am really confused that you are using another paper's work to argue the results in this work. I tried to read another paper. It seems that most of the contents regarding the analysis of convergence in NTK are identical, with difference of ignoring the results about the high probably argument and generalization performance analysis. I think the larger step size is, the faster the rate is (just as shown in Theorem 3.4.), right? What does this bound mean?
> > > > > >
> > > > > > The most important question is where the proof of Theorem 3.4. B.2. is NOT complete. It just gives the expression of the gradient in terms of G, right? I guess you would still need to use another paper's derivations, e.g., [1].
> > > > > >
> > > > > > Why does $\sum^K_{k'k}T_{k'k}=1$ make the convergence independently on $\xi$? Note that you did not provide the relation between $T$ and the rate. How to plug in this result to Theorem 3.4.?

---

> > > > > > > ### Author Response · Authors · 2022-11-27
> > > > > > > **3rd round response - Part I**
> > > > > > >
> > > > > > > Dear Reviewer N8Hr,
> > > > > > >
> > > > > > > Thanks for your super prompt feedback, and we are very thankful for your time. We are delighted to further explain your remaining concerns.
> > > > > > >
> > > > > > > > Theorem 3.4 should be true all $t$, right?... I think the larger step size is, the faster the rate is (just as shown in Theorem 3.4.), right?
> > > > > > >
> > > > > > > We will try our best to understand your confusion. The relationship that $t$ is in the order of inversely proportional to $\eta$ only holds if our analysis and assumptions meet the standard of over-parameterized networks in the NTK regime, where the key observation is that weights change lazily and we have to bound the model updates. We affirm that Theorem 3.4 should be true for all $t$ *if* $\eta$ is properly selected to ensure a slow change of model weights. Since local model updates are bounded, the algorithm requires more communication cost when $t=1$, which could be expensive under FL.
> > > > > > >
> > > > > > > To provide more intuition on the convergence of NTK, we need to meet both small model weights movement and proper selections for $\eta$ and $t$ to achieve error $\epsilon$ with probability $1-\delta$. Specifically, in the overparameterized regime, we require the update step to be small enough that a derivative on model weights over time can be approximated by gradients of loss, and the loss always decreases as we run gradient descent. These are essential steps in NTK analysis and the basic derivations in the NTK literature [1,2].
> > > > > > >
> > > > > > > Essentially, in this work, we would like to highlight that we focus on the convergence affected by $\lambda$ (the smallest eigenvalue of FL’s Gram matrix) and how the proposed loss in FedMT can affect $\lambda$, thus affecting convergence in FL.
> > > > > > >
> > > > > > > > The most important question is where the proof of Theorem 3.4. B.2. is NOT complete. It just gives the expression of the gradient in terms of G, right? I guess you would still need to use another paper's derivations, e.g., [1].
> > > > > > >
> > > > > > > As we stated in our manuscript, our convergence results in Theorem 3.4 were extended from [1], a recent convergence analysis tool for FL. Therefore, for simplicity, we pointed the reviewer to [1] for the lemmas used in both works. In our work, we used some conclusions and techniques from [1] and intentionally omitted their detailed derivation because we hoped to clearly highlight our contributions and novelty while giving proper credit to [1] that we refer to, rather than confusing the readers.
> > > > > > >
> > > > > > > Yes, Theorem 3.4’s convergence results can be easily derived from B.2 by trivially replacing our Gram matrix with [1]. To this end, we derived that each element Gram matrix using FedMT loss correction is just a weighted version of the one without loss correction. Specifically, using matrix format, our gramma matrix $\Lambda H \Lambda$, where $\Lambda$ is a diagonal matrix whose elements are either $(\sum_{j} T_{jk}^{-1})^{1/2}$ or $(\sum_{j} Q_{jk}^{-1})^{1/2}$, and $H$ follows the same form of the Gram matrix in [1]. It is easy to see that after replacing the gram matrix in [1] with this weighted version and following the same derivative, there is a constant scale factor to its bound in the proof for convergence. We are delighted to repeat the same details that we borrowed from [1] in the final version.
> > > > > > >
> > > > > > > > I tried to read another paper. It seems that most of the contents regarding the analysis of convergence in NTK are identical, with difference of ignoring the results about the high probably argument and generalization performance analysis.
> > > > > > >
> > > > > > > Comparing with the tool [1] for vanilla FedAvg analysis, we made the following contributions towards the convergence analysis results for FedMT:
> > > > > > >
> > > > > > > 1. First, [1] does a convergence analysis on FedAvg, but in our FedMT algorithm, we used different losses. Then, as we replied to the previous question, we showed the Gram matrix of FedMT with label projection, which is nicely expressed as a weighted version of that in [1].
> > > > > > >
> > > > > > > 2. Second, note that [1] is for a regression problem and did not consider multiple classes. Thus, we extended its analysis to multiple classes and derived the blocked Gram matrices representations.
> > > > > > >
> > > > > > > 3. Third, we further present Corollaries 3.5 and 3.6 to show how the factors in FedMT algorithms affect convergence.
> > > > > > >
> > > > > > > References:
> > > > > > >
> > > > > > > [1] FL-NTK: A Neural Tangent Kernel-based Framework for Federated Learning Convergence Analysis. (ICML 2021)
> > > > > > >
> > > > > > > [2]  On Exact Computation with an Infinitely Wide Neural Net. (NeurIPS 2019)

---

> > > > > > > > ### Author Response · Authors · 2022-11-27
> > > > > > > > **3rd round response - Part II**
> > > > > > > >
> > > > > > > > > Why does $\sum^K_{k'k}T^{-1}_{k'k}=1$ make the convergence independently on $\xi$? Note that you did not provide the relation between $T$ and the rate. How to plug in this result to Theorem 3.4.?
> > > > > > > >
> > > > > > > > We connected the relationship $T$ with convergence by the Gram matrix $G$ of FedMT calculated in our appendix B.2. Let us explain how we related $T$ and $\xi$ to convergence step by step. Note T reflects $\xi$. According to the NTK theorem, which is also shown in our Theorem 3.4, the convergence rate is dominated by the smallest eigenvalue of the Gram matrix. $G$. When $\sum^K_{k'k}T^{-1}_{k'k}=1$, it is easy to see from the form of G keep as we showed in B.2 keeps the same regardless of $\xi$. Thus, its smallest eigenvalue, $\lambda$ does not change when xi changes. Then we plug in this to Theorem 3.4 with other parameters fixed, the NTK convergence does not depend on $\xi$ in this case.
> > > > > > > >
> > > > > > > > We are grateful that you (reviewer N8Hr) asked the questions so that we have this valuable opportunity to provide more details of our work on the Openreview platform to help readers understand our theoretical derivations and results better. We have well received your kind messages that we can include more details in our appendix and will be very happy to improve this part based on your questions. If you have any remaining questions, please don't hesitate to let us know. If our responses could have addressed your concerns or some of your concerns, we would appreciate it if you could re-evaluate our score.

---

> > > > > > > > > ### Comment · Area_Chair_5emT · 2022-12-01
> > > > > > > > > **I think we need contextualize this discussion better.**
> > > > > > > > >
> > > > > > > > > Hi Reviewer N8Hr, authors,
> > > > > > > > >
> > > > > > > > > I have been watching this thread of discussions as this paper falls within the "borderline" range as defined by ICLR.
> > > > > > > > >
> > > > > > > > > While I enjoyed watching the discussion (I personally work on NTK too), from the paper I feel this NTK theorem isn't the main-seller contribution, instead more a "supplemental" contribution that helps understand authors' findings. Seemingly, the major motivation, idea, and experimental results can go rather independently of NTK. I'd love to learn from both sides:
> > > > > > > > > - To authors: do you feel it more appropriate to move the NTK analysis into supplementary, avoiding it distracting readers' focus?
> > > > > > > > > - To Reviewer N8Hr: if taking NTK part out of your consideration, how would you evaluate the remaining main contributions (methods, and results) - would you consider it more passing the bar? From your previous comments it seems you feel positively, quoting: "I agree that the proposed method is novel in the sense that it considers a practical learning issue (mixed-type labels) in the FL setting."
> > > > > > > > >
> > > > > > > > > I hope we could reduce opinion divergence effectively by digging this way.
> > > > > > > > >
> > > > > > > > > AC

---

> > > > > > > > > > ### Author Response · Authors · 2022-12-02
> > > > > > > > > > **contextualize this discussion**
> > > > > > > > > >
> > > > > > > > > > Dear AC, thank you for recognizing our main contribution in this paper and valuable suggestions. We are delighted to move the NTK convergence (Theorem 3.4) to the appendix and add the existing shown in our supplementary material to the main text and other necessary details to highlight our contribution.  We also thank reviewer N8Hr for the comment. We reply to the reviewer’s comments as follows.
> > > > > > > > > >
> > > > > > > > > > - Regarding reviewer N8Hr’s question on eigenvalues, it is easy to see that if all the loss-corrected weights are positive, the smallest eigenvalue is strictly positive. In both cases we discussed in Corollary 3.5 and Corollary 3.6, we have $\sum_{k’} Q_{kk’}^{-1}>0$ and $\sum_{k’} T_{kk’}^{-1}=1>0$, which is easy to see that the smallest eigenvalues in these two cases must be positive. The two cases we discussed are proper abstractions from some practical settings. For more general scenarios, [1] has shown the techniques on NTK analysis w.r.t. kernel matrix that is not perfectly positive definite. It is not our focus to investigate the explicit form of $\mathbb{T}^{-1}$ as we consider it as a reconstructable to $\mathbb{T}$ following [3]. To get positive definite reconstructable $\mathbb{T}^{-1}$, this part of the technique follows the existing loss correction literature, e,g., by building a non-negative risk estimator, by eliminating negative terms [2]. Last, in practice, we’ve also shown the proposed FedMT converges well even just using the pseudo inverse.
> > > > > > > > > > [1] FL-NTK: A Neural Tangent Kernel-based Framework for Federated Learning Convergence Analysis. ICML 2021
> > > > > > > > > > [2] Positive-unlabeled Learning with Non-negative Risk Estimator. NeurlPS 2017
> > > > > > > > > > [3] A theory of learning with corrupted labels. JMLR 2017
> > > > > > > > > > - Regarding the concern on “the algorithm is almost identical to FedAvg,” indeed FedMT is easy to implement, yet sheds new light on the setting of mixed type labels in FL. **building FedMT on FedAvg is our merit** as FedAvg is a fundamental FL method and serves as the base of many existing FL methods (e.g., FedProx, FedBN, Scaffold, etc.). Thus, we intentionally chose FedAvg as a backbone framework to demonstrate the feasibility of our strategy rather than other more advanced methods. We stated in our conclusion that “As a plug-and-play method, FedMT can be applied to the non-IID setting by combining with advanced FL schemes, including” and reviewer Xv7p also recognized our merit of “can be easily combined with various FL methods”.
> > > > > > > > > > - Regarding the use of NTK analysis for optimization. As the AC pointed out, “this NTK theorem isn't the main-seller contribution, instead more a "supplemental" contribution that helps understand authors' findings “, the limitation of NTK is not the focus of our work. The NTK is widely used in optimization [1-5], and it inspires many works in recent ML venues.
> > > > > > > > > > [1] Jacot, A., Gabriel, F., & Hongler, C. (2018). Neural tangent kernel: Convergence and generalization in neural networks. NeurlPS
> > > > > > > > > > [2] Allen-Zhu, Z., Li, Y., & Song, Z. (2019). On the convergence rate of training recurrent neural networks. NeurlPS
> > > > > > > > > > [3] Allen-Zhu, Z., Li, Y., & Song, Z. (2019). A convergence theory for deep learning via over-parameterization. ICML
> > > > > > > > > > [4] Lee, J., Xiao, L., Schoenholz, S., et al. (2019). Wide neural networks of any depth evolve as linear models under gradient descent. NeurlPS 2019
> > > > > > > > > > [5] Dukler, Y., Gu, Q., & Montúfar, G. (2020). Optimization theory for RELU neural networks trained with normalization layers. ICML
> > > > > > > > > > [6] Arora, S., Du, S. S., Hu, W., et al. (2019). On exact computation with an infinitely wide neural net. NeurlPS
> > > > > > > > > > - Regarding the epochs, as we replied earlier in round 1 of the rebuttal, we plotted the first 50 epochs in Figure 2 to highlight the difference in the effect of different parameters of our interest. We’ve shown all the 100 training epochs in Appendix B.1 and our code is available.
> > > > > > > > > > - Regarding the comments on revision, we do not think the revision according to AC’s suggestion is a major revision as the reviewer’s major concern is around the NTK framework itself in Theorem 3.4, which is not the key focus of this work. During the rebuttal, we mainly point the reviewers to the existing narratives in our submission and literature for clarification. We are happy to follow the AC’s recommendations to make proper editing while addressing reviewer N8Hr’s concern.
> > > > > > > > > > We thank reviewer N8Hr again for the discussion.
> > > > > > > > > >
> > > > > > > > > > ---
> > > > > > > > > > We appreciate AC and all the reviewers’ valuable time and encouraging comments again. We hope to bring this important and practical mix-type labels problem to the FL community for the first time. As the initial step towards investigating the novel problem, we propose a versatile method, FedMT, together with our insights into method design inspired by theory. Our experiments validate the key factors in mix-type label FL. We believe our work can be a good benchmark to inspire future follow-up work in this direction.

---

> > > > > > > > > > > ### Comment · Reviewer_N8Hr · 2022-12-02
> > > > > > > > > > > **The proof is wrong!**
> > > > > > > > > > >
> > > > > > > > > > > How to characterize "all the loss-corrected weights are positive"? I never saw this condition in both main text and supplement. Please define the corrected weights and show how this condition can lead to a strictly positive matrix $G$. $T$ and $Q$ are both shown in the matrix $G$. Besides, how easy ("which is easy to see") to link these two properties to the final convergence rate? I did not find it in the supplement and rebuttal. A correct proof is needed, otherwise, the statement of the theorem and contributions of this work are not consistent with the technical proof.
> > > > > > > > > > >
> > > > > > > > > > > Again, the quality of the presentation of this paper is not good. The expression of $T$ in Corollary 3.6 is not consistent with the one shown in proof (please see $\xi$, sometimes it is in the numerator while sometimes it appeared in the denominator). Also, the notation $T$ is not consistent again. I can understand these are all typos again. But, there are too many typos in this paper.
> > > > > > > > > > >
> > > > > > > > > > > It is good to see the advantages of this method numerically. If there is no strong technical result, sufficient numerical results are needed. So far, all the results are mainly tested on the CIFAR 100 and MDSUPDRS19, for example, how about the performance on ImageNet. Also, the performance of FedMT is the not best (I can understand there is another typo there, which authors did not mark the best one with boldface as promised.) Will this happen for a larger number of classes? Because I found that when $n$ is small, FedMT performs well, e.g., MDSUPDRS19. Even convergence curves for both training and testing in comparison of all the tested algorithms/methods are not plotted. They are very basic figures, which are unfortunately not shown in either main text or supplement.
> > > > > > > > > > >
> > > > > > > > > > > Overall, there are major concerns left for both theory and experiments. The current paper is not ready for acceptance at this stage.

---

> > > > > > > > > > > > ### Author Response · Authors · 2022-12-04
> > > > > > > > > > > > **Response**
> > > > > > > > > > > >
> > > > > > > > > > > > We would like to thank you again for your time and comments.
> > > > > > > > > > > >
> > > > > > > > > > > > > Disagreement on the theory.
> > > > > > > > > > > >
> > > > > > > > > > > > The high-level idea is to embed the transition matrix to the existing NTK framework to understand the connection of FedMT with loss correction with the classical FL. The key takeaway from the theoretical analysis is that the loss reweighting yields in a Gram matrix $\tilde{H} = \Gamma H \Gamma$ where $\Gamma$ is a diagonal matrix consisting of the weights associated with loss reweighting and $H$ is the gram matrix of classical FL, i.e., FedAvg. The property and the convergence associated with $H$ and the technique to obtain a positive definite approximation to H was presented in existing literature (Huang et al.). Following a similar analysis on $H$, we can show $\Gamma H \Gamma$ with positive weights of $\Gamma$ weights is positive definite (corollary 3.5 and corollary 3.6). We will accommodate the AC and reviewers’ comments to more properly state our theoretical analysis, which aims to motivate the design and help understand some properties of FedMT, rather than the NTK framework itself.
> > > > > > > > > > > >
> > > > > > > > > > > >
> > > > > > > > > > > >
> > > > > > > > > > > > > The expression of $T$ in Corollary 3.6 is not consistent with the one shown in proof (please see $\xi$, sometimes it is in the numerator while sometimes it appeared in the denominator).
> > > > > > > > > > > >
> > > > > > > > > > > > We believe we have clarified the notation confusion. We will carefully proofread our manuscript for the final version. We are thankful for reviewer N8Hr's understanding.
> > > > > > > > > > > >
> > > > > > > > > > > >
> > > > > > > > > > > >
> > > > > > > > > > > > > how about the performance on ImageNet.
> > > > > > > > > > > >
> > > > > > > > > > > > As we earlier responded to reviewer xV7p in this [post](https://openreview.net/forum?id=lCzuxqRbThP&noteId=9ROYS0pJ-jZ), we did not run the experiment on this dataset since ImageNet only has one label space and there does not exist commonly used criteria for creating different label spaces. Therefore, there could be some disagreement on the number of classes in a label space and how these classes are chosen. Hence, we believe the comparison on ImageNet could bring up confusion and disagreement and do not run the experiment on ImageNet. Nevertheless, we added more experiment results, including [with different backbones](https://openreview.net/forum?id=lCzuxqRbThP&noteId=9ROYS0pJ-jZ), [robustness of estimating $T$](https://openreview.net/forum?id=lCzuxqRbThP&noteId=kn9wxeuoOX7).
> > > > > > > > > > > >
> > > > > > > > > > > > > Also, the performance of FedMT is the not best (I can understand there is another typo there, which authors did not mark the best one with boldface as promised.)
> > > > > > > > > > > >
> > > > > > > > > > > > We guess the reviewer is referring to experiment results in the last line of Table 1 (FedMatch: 30.71(0.59), Our: 30.63(0.06) ). We did not boldface the two methods intentionally as these two methods have comparable performances. We clearly mentioned on page 8 of the main text “We achieve similar but much more stable performance compared with FedMatch when $n = 25$”. As the first work tackling the mixed-type label problem, we did not aim for the best-ever solution. We believe our solution can be a basline and inspire future work. Lastly, we highlighted in our manuscript that “Future work could combine FedMT with similar approaches used in FedMatch...” and our method is versatile “As a plug-and-play method, FedMT can be applied to non-IID setting by combining with advanced FL schemes...”
> > > > > > > > > > > >
> > > > > > > > > > > >
> > > > > > > > > > > >
> > > > > > > > > > > > > Even convergence curves for both training and testing in comparison of all the tested algorithms/methods are not plotted. They are very basic figures, which are unfortunately not shown in either main text or supplement.
> > > > > > > > > > > >
> > > > > > > > > > > > We ran repeated experiments with many comparison methods under a lot of different settings. We, therefore, did not plot the convergence curve for every single experiment. With more preparation time, we are very happy to add these plots to the final version.

---

> > > > > > > > ### Comment · Reviewer_N8Hr · 2022-11-28
> > > > > > > > **How to ensure the positivity of the smallest eigenvalue of G?**
> > > > > > > >
> > > > > > > > The key difference between this work and [1] would be the weight matrices $T$ and $Q$. Based on the closed form expression of the Gram matrix shown in page 16, it seems that $G$ is weighted by both $T$ and $Q$ everywhere in the sense that for each element of $G$ it will be weighted by an entry of either $Q^{-1}$ or $T^{-1}$. After this kind of re-weighting, how will the smallest eigenvalue of $G$ be changed? is there any condition on $T$ and $Q$ so that $\lambda$ is always positive. Otherwise, there convergence rate results is not correct, right? You mentioned in the response, the convergence rate holds for general $T$. I doubt it is true. It will be great if you can show $\lambda>0$. Thanks.

---

### Decision · Program_Chairs · 2023-01-20

**Decision:**

Reject

**Justification For Why Not Higher Score:**

The paper has certain promise in its settings and experimental results. But one consensus is that the theoretical contribution is thin in this paper. It should be mainly appreciated as a simple, empirical algorithm in an unexplored setting. More outstanding issues include the confusing description and non-trivial estimation of matrices Q and T, as well as the lack of discussion about the combination with more advanced FL algorithms. The paper's clarity, as well as self-consistency, also needs more revision work. Therefore, AC recommends the rejection.

**Justification For Why Not Lower Score:**

N/A

**Metareview: Summary, Strengths And Weaknesses:**

This paper aims to tackle a novel practical FL setting with mixed-type label classification (e.g., disease diagnosis where clients use different label spaces). As the first work, the authors propose a versatile method, FedMT, along hew insights into method design inspired by theory. Their experiments validate the key factors and show better performance than alternative solutions.

Main strengths are:
- Novel setting, definitely bringing some new ideas to the FL field
- Simple algorithm (a variant of FedAVG) based on projections, which are backed by NTK theory-derived proofs
- Consistent gains over experiments
- Code is available and details are included in the appendix.

Main critiques are:
- The setting might be a bit artificial
- Limited novelty w.r.t. (Rooyen & Williamson, 2017)
- Clarity issue of two projections; lots of typos in the original submission
- Theory is very limited (due to NTK-based), and seems detached from practice
- Improvements are often marginal

**Summary Of Ac-Reviewer Meeting:**

Due to the large score variation after rebuttal, an AC-reviewer Meeting was held in Dec 7. All four reviewers attended. Notes are summarized below:

- Reviewers B327 and xV7p appreciated the empirical elegance and effectiveness of this work. They had some questions regarding the theory's assumptions, which got clarified by the authors. One remaining issue, as they pointed out, is that the two projects terms are confusing in notations and need to be revised more.

- Reviewer 23rr concurred that obtaining T and Q is in general not an easy task and should be clarified in the revision. The theory is slightly detached to the empirical practice, to support two projections.  Also noted is that this algorithm was built mainly on top of FedAvg and it was unclear how it will apply to other Fed optimizers.
Furthermore, the reviewer suggested that the authors borrow ideas from (Rooyen & Williamson, 2017) loss correction, which limited their novelty. Authors explained in rebuttal that (Rooyen & Williamson, 2017)  studies label corruption issues using projection matrices for loss correction while this paper tackles the mixed-type label issue under the FL supervised learning; and FL brings several unique challenges. The reviewer thinks the authors' rebuttal "kind of" addresses the notation clarity issue.

- Reviewer N8Hr, the most critical one, has his/her major complaints on many typos in proofs/notations. Indeed, thorough proofreading and revision would be demanded.
He/she also suggested NTK cannot accurately characterize NN optimization under realistic width; yet during AC-reviewer meeting, a consensus is reached that we shouldn't penalize the authors for using an off-the-shelf and popular theory tool, while the theory tool itself has inherent limitations. So it's agreed by AC and reviewers that the NTK concern was resolved.
Theorem 3.4 proof could be more detailed in explaining why PSD. It seems obvious when all columns are centered, but what if all columns are from heterogeneous data sources and hence columns are not centered?
Lastly, the reviewer suggested to add more convergence plot (w.r.t epoch) and error bars, to promote the paper's reproducibility. This concern was also alleviated by the authors' provided codes.